# Assessing cardiovascular disease risk and social determinants of health: A comparative analysis of five risk estimation instruments using data from the Eastern Caribbean Health Outcomes Research Network

Jeremy I. Schwartz[1]*, Christina Howitt[2], Sumitha Raman[3], Sanya Nair[4], Saria Hassan[5], Carol Oladele[1], Ian R. Hambleton[2], Daniel F. Sarpong[6], Oswald P. Adams[7], Rohan G. Maharaj[8], Cruz M. Nazario[9], Maxine Nunez[10], Marcella Nunez-Smith[1]

1 Equity Research and Innovation Center, Section of General Internal Medicine, Yale School of Medicine, New Haven, Connecticut, United States of America, 2 George Alleyne Chronic Disease Research Centre, Caribbean Institute for Health Research, The University of the West Indies, Barbados, 3 Division of General Internal Medicine, George Washington University School of Medicine and Health Sciences, Washington, DC, United States of America, 4 Yale University, New Haven, Connecticut, United States of America, 5 Division of General Internal Medicine, Emory Department of Medicine, Emory University School of Medicine, Atlanta, Georgia, United States of America, 6 Section of General Internal Medicine and Office of Health Equity Research, Yale School of Medicine, New Haven, Connecticut, United States of America, 7 Department of Family Medicine, Faculty of Medical Sciences, University of the West Indies Cave Hill, Cave Hill, Barbados, 8 Department of Paraclinical Sciences, University of the West Indies, Saint Augustine, Trinidad and Tobago, 9 Department of Biostatistics and Epidemiology, Graduate School of Public Health, University of Puerto Rico at Medical Sciences Campus, San Juan, Puerto Rico, 10 School of Nursing, University of the Virgin Islands, St. Thomas, US Virgin Islands

* Jeremy.schwartz@yale.edu

## Abstract

### Background

Accurate assessment of cardiovascular disease (CVD) risk is crucial for effective prevention and resource allocation. However, few CVD risk estimation tools consider social determinants of health (SDoH), despite their known impact on CVD risk. We aimed to estimate 10-year CVD risk in the Eastern Caribbean Health Outcomes Research Network Cohort Study (ECS) across multiple risk estimation instruments and assess the association between SDoH and CVD risk.

### Methods

Five widely used CVD risk estimation tools (Framingham and WHO laboratory, both laboratory and non-laboratory-based, and ASCVD) were applied using data from ECS participants aged 40–74 without a history of CVD. SDoH variables included educational attainment, occupational status, household food security, and perceived social status. Multivariable logistic regression models were used to compare differences in the association between selected SDoH and high CVD risk according to the five instruments.

**Data Availability Statement:** Data are publicly accessible through an application process by the ECHORN Data and Scientific Advisory Committee (DASR) at https://www.echorn.org/request-echorn-data. Individual level participant data and data dictionaries are made available via DASR request. Aggregate level data is made available via ExploreECHORN. All other granular data as well as additional, related documents are made available via DASR request. ECHORN wave 1 data is available now via DASR and ExploreECHORN, ECHORN wave 2 data will be made available by those same mechanisms after all sites have closed study activities.

**Funding:** NIH grants U24MD006938, U54MD010711, K23HL152368, UL1TR000142, and Yale School of Medicine.

**Competing interests:** The authors have declared that no competing interests exist.

## Findings

Among 1,777 adult participants, estimated 10-year CVD risk varied substantially across tools. Framingham non-lab and ASCVD demonstrated strong agreement in categorizing participants as high risk. Framingham non-lab categorized the greatest percentage as high risk, followed by Framingham lab, ASCVD, WHO lab, and WHO non-lab. Fifteen times more people were classified as high risk by Framingham non-lab compared with WHO non-lab (31% vs 2%). Mean estimated 10-year risk in the sample was over 2.5 times higher using Framingham non-lab vs WHO non-lab (17.3% vs 6.6%). We found associations between food insecurity, those with the lowest level compared to the highest level of education, and non-professional occupation and increased estimated CVD risk.

## Interpretation

Our findings highlight significant discrepancies in CVD risk estimation across tools and underscore the potential impact of incorporating SDoH into risk assessment. Further research is needed to validate and refine existing risk tools, particularly in ethnically diverse populations and resource-constrained settings, and to develop race- and ethnicity-free risk estimation models that consider SDoH.

## Introduction

Cardiovascular disease (CVD) is the leading cause of mortality worldwide and is responsible for 28% of global deaths [1]. In the islands of the Caribbean, the prevalence of CVD risk factors, such as diabetes and hypertension, is substantial and growing and confers a high societal economic cost [2–4]. A widely adopted approach to individualized and population-based CVD prevention is the assessment of 10-year CVD risk [1]. It has been suggested that treatment of patients based on their individualized CVD risk is more prudent than basing treatment decisions on specific risk factors, such as hypertension and hyperlipidemia, alone [5]. Estimating the risk of cardiovascular events has, thus, become a cornerstone of clinical practice as it influences clinician recommendations for lifestyle and pharmacologic primary prevention. That said, controversy still exists as to whether this approach actually improves clinical outcomes [6]. Additionally, from the perspective of population health, these estimations allow for targeted approaches to prevention messaging, health sector planning, and resource allocation [7].

Numerous multivariable risk estimate tools have been developed to estimate the 10-year risk of developing a cardiovascular event [8]. The accuracy and applicability of these tools are linked to the populations with and for whom they were developed and validated. Studies have shown that these tools can underestimate or overestimate [9] actual CVD risk and can have limited accuracy in certain populations [10]. Many of the tools were developed in and validated among, ethnically and sociodemographically homogeneous populations, typically in high-income country settings, limiting their applicability to individuals of different socio-economic statuses and multi-ethnic populations [11]. That said, efforts have been made to broaden their applicability, through validation in multi-ethnic cohorts and adaptations to omit required laboratory parameters, which may be difficult to obtain in some settings [10, 11]. In 2007, the World Health Organization (WHO) and the International Society of Hypertension published the first series of CVD risk charts developed specifically for LMIC, with a different chart for each of the 14 WHO sub-regions [9]. These charts were updated in 2019,

with a different chart corresponding to each of the 21 Global Burden of Disease regions, including one for the Caribbean that was validated using data from Trinidad and Barbados [1].

Though these newer tools incorporate data from more diverse populations, few have incorporated social determinants of health (SDoH), despite the well-documented associations between certain SDoH and CVD. Numerous studies have shown positive associations between SDoH and CVD risk, including household food security, educational attainment, and other markers of socioeconomic status [12–14]. For example, in some settings, food insecurity has been linked to higher prevalence and poor management of CVD risk factors like obesity, diabetes, and hypertension [15]. Lower educational attainment and non-professional occupations have also been associated with higher CVD incidence and mortality across studies [12]. Given these associations, incorporating social determinants of health into CVD risk estimate tools could improve their accuracy across, and generalizability to, diverse populations and settings [16]. However, this landscape and its evidence base continue to evolve. A recent study demonstrated that the removal of race and the addition of SDoH neither improved nor worsened performance of the ASCVD instrument [17]. Concurrently, the American Heart Association released an updated version of ASCVD, called PREVENT, that both removed race and added a measure of SDoH [18].

This study aims to determine the estimated 10-year CVD risk range in the Eastern Caribbean Health Outcomes Research Network (ECHORN) Cohort Study (ECS) using multiple widely used risk estimate tools, including the 2019 Caribbean WHO risk chart. Furthermore, we seek to determine how the association of selected SDoH—educational attainment, occupational status, household food security, and perceived social status—with CVD risk varies by instrument.

## Methods

### Overview of study design and data collection

The study design and data collection process have been documented in detail elsewhere [19]. As previously described, ECS utilized a multistage probability sampling design in three of the four sites (Barbados, Puerto Rico, and Trinidad) and simple random sampling in the fourth site, US Virgin Islands (St. Thomas and St. Croix). Island-wide samples were obtained from the US Virgin Islands and Barbados, while the larger islands of Puerto Rico and Trinidad selected two communities with similar demographics (similar distributions of age, race/ethnicity, sex, and educational levels) representative of the the general island population. Participants were considered eligible if they were non-institutionalized, aged over 40 years, English or Spanish speaking, had reliable contact information, and were semi-permanent or permanent residents of the site for the past 10 years, with no plans to relocate in the next 5 years. Pregnant women were excluded.

Baseline data collection occurred between the following dates at each site: US Virgin Islands (June 5, 2013-October 28, 2015), Barbados (November 5, 2013-May 12, 2016), Trinidad (May 6, 2014-June 12, 2018), and Puerto Rico (May 29, 2014-June 12, 2018). Participants completed questionnaires that captured sociodemographic information, health status, and health behavior information. The survey was conducted using computer-guided and audio-assisted software. Physical measurements were taken, including height, weight, waist, hip, and neck circumference, and blood pressure. A fasting venous blood sample was collected for glucose, HDL, LDL and total cholesterol. A full dictionary of collected variables is publicly available on the Explore ECHORN online platform [20].

## Measures used in analyses

In our analyses, we included participants aged 40–74 years, based on the common age range across the five selected CVD risk instruments (see below) and without a known history of CVD at baseline. Data were analyzed using the Stata software package (version 16, StataCorp, College Station, Texas).

Variables included in the calculation of CVD risk varied across tools. In this section, all the variables used across all five risk calculation tools are described. Age was obtained by ECS participant self-report and operationalized as means and 10-year categories in analyses. Sex, race, educational attainment, and occupation were obtained by participant self-report.

Clinical and lifestyle variables examined included current smoking and use of antihypertensive medications (obtained by self-report), body mass index (obtained by measurement of height and weight), HDL and total cholesterol (obtained by laboratory testing), and hypertension and diabetes (described below). Smoking was defined as current smoking of at least 20 cigarettes or 1 cigar or half an ounce sachet of loose tobacco per month. Participants were considered to have diabetes if they met either of the following criteria: a fasting plasma glucose measurement of 126 mg/dL or higher or a self-reported prior diagnosis [2]. Hypertension was defined as either a systolic blood pressure ≥140 mmHg or a diastolic blood pressure ≥90 mmHg or self-reported current use of antihypertensive medication. These definitions are consistent with those used in population-based surveys in the US (NHANES) and globally (WHO STEPS) [21–23].

In terms of SDoH, household food security was assessed by the Latin American and Caribbean Food Security (ELCSA) scale [24]. Responses were categorized as follows: 'household food secure' (score = 0), 'mild household food insecurity' (score = 1–3), 'moderate household food insecurity' (score = 4–6), 'severe household food insecurity' (score = 7–9). Social status was scored 1–10 based on the following question: "Look at this figure with steps numbered 1 at the bottom to 10 at the top. If the ladder represents the richest people of this island and the bottom represents the poorest people of this island, on what number step would you place yourself?" [25]. Data were categorized according to tertile. Educational attainment was categorized as less than high school, high school graduate, some college, and college and beyond. Self-reported occupation was collapsed into three categories: professional, semi-professional, and non-professional. Analyses looking at associations of SDoH and CVD risk were adjusted for age and gender.

Ten-year risk of cardiovascular events for each individual was calculated using five widely used CVD risk estimate tools: Framingham Risk Score laboratory-based (*Framingham lab*) [26], Framingham Risk Score non-laboratory-based (*Framingham non-lab*) [26], American College of Cardiology/American Heart Association Pooled Cohort Risk equations (*ASCVD*) [27], World Health Organization 2019 laboratory-based risk charts (*WHO lab*) [1] and World Health Organization 2019 non-laboratory-based risk charts (*WHO non-lab*) [1]. The salient features of each risk calculator are summarized in Table 1. Framingham lab, ASCVD, and WHO lab include age, sex, systolic blood pressure, treatment of hypertension, total cholesterol levels, current smoking, and history of diabetes mellitus. Framingham lab and ASCVD also include high density lipoprotein (HDL) and ASCVD includes race ("African American" or "Other"; Though the online tools allow for selection of "White", the underlying algorithm groups "White" and "Other" together.) For Framingham non-lab and WHO non-lab, lipid measurements are replaced with body mass index (BMI). We used the Caribbean-specific risk charts for the WHO risk estimates (Table 1). The ASCVD tool requires that race be designated as either "African American" or "Other". Our questionnaire allowed the following responses to the question "To which racial or ethnic group or groups would you say you belong? (Check

**Table 1. Salient features of each of the five CVD risk estimate instruments used in this study.**

| | Framingham (lab/non-lab) | ASCVD | WHO (lab/non-lab) |
|---|---|---|---|
| **Population** | Derived from the Framingham cohort study, with 8491 individuals | Derived from several cohort studies, including Framingham, with 21 985 individuals | Derived from 85 prospective cohorts with 376 177 individuals |
| | Age range: 30–74 yrs | Age range: 40–79 years | Age range: 40–80 yrs |
| | Location: North America | Location: North America | Location: Europe, North America, Japan, Australia |
| **Risk factors** | Lab: age, sex, smoking, systolic blood pressure, blood pressure treatment, total cholesterol, HDL cholesterol, diabetes | Age, sex, smoking, systolic blood pressure, blood pressure treatment, total cholesterol, HDL cholesterol, diabetes, race | Lab: age, sex, smoking, systolic blood pressure, total cholesterol, diabetes |
| | Non-lab: Age, sex, smoking, systolic blood pressure, blood pressure treatment, diabetes, BMI | | Non-lab: Age, sex, smoking, systolic blood pressure, BMI |
| **Timeframe & outcomes** | 10-yr risk of fatal and nonfatal CVD (coronary heart disease, stroke, peripheral artery disease, or heart failure) | 10-yr risk of fatal and nonfatal CVD (coronary heart disease or stroke) | 10-yr risk of fatal and nonfatal CVD (coronary heart disease or stroke) |
| **Statistical model** | Cox survival models | Cox survival models | Cox survival models |
| **Are country-specific versions available?** | No | No | Yes—different charts for 21 regions worldwide. Region-specific calibrations were based on regional incidence of risk factors. |

all that apply)”: white, black or African, Caribbean, Asian (for example Japanese, Chinese, Laotian, Thai, Pakistani or Cambodian), East Indian, Hispanic or Latino, Mixed or multi-racial, Puerto Rican or Boricua, or other. Participants who either selected “Black or African” or “Caribbean” were classified as “African American”. All others were classified as “Other”.

There is some heterogeneity in clinical endpoints between these tools that is worth noting. The clinical endpoints for the ASCVD and WHO tools are fatal and non-fatal myocardial infarction and stroke, whereas the Framingham tools endpoints also include angina, heart failure, transient ischemic attack, and peripheral artery disease.

To achieve consistency across the five risk estimate tools, we categorized 10-year risk as low, intermediate, and high. For the Framingham tools, we used the recommended risk categorizations: low risk (<10%); intermediate risk (10% to <20%); high risk (≥20%). For ASCVD, we combined the low risk (<5%) and borderline risk (5% to <7.5%) categories into a single low risk category and retained the intermediate risk (≥7.5% to <20%) and high risk (≥20%) categories. For the WHO tools, we combined the very low risk (<5%) and low risk (5% to <10%) categories into a single low risk category, retained the intermediate risk (10% to <20%) category, and combined the high risk (20% to <30%) and very high risk (≥ 30%) categories into a single high risk category.

We excluded participants with missing data for any of the variables required by the five tools to ensure that the same participants were used to compute all scores (Fig 1). We provide sensitivity analyses to compare demographics among included and excluded participants (S1 Table).

## Ethics approvals

The ECS was approved by the Yale University Human Subjects Investigation Committee and the Institutional Review Boards of the University of the West Indies, the Ministry of Health Trinidad and Tobago, the University of Puerto Rico, Medical Sciences Campus and the University of the US Virgin Islands. Written informed was obtained from study participants. The approvals also included the use and analysis of deidentified data.

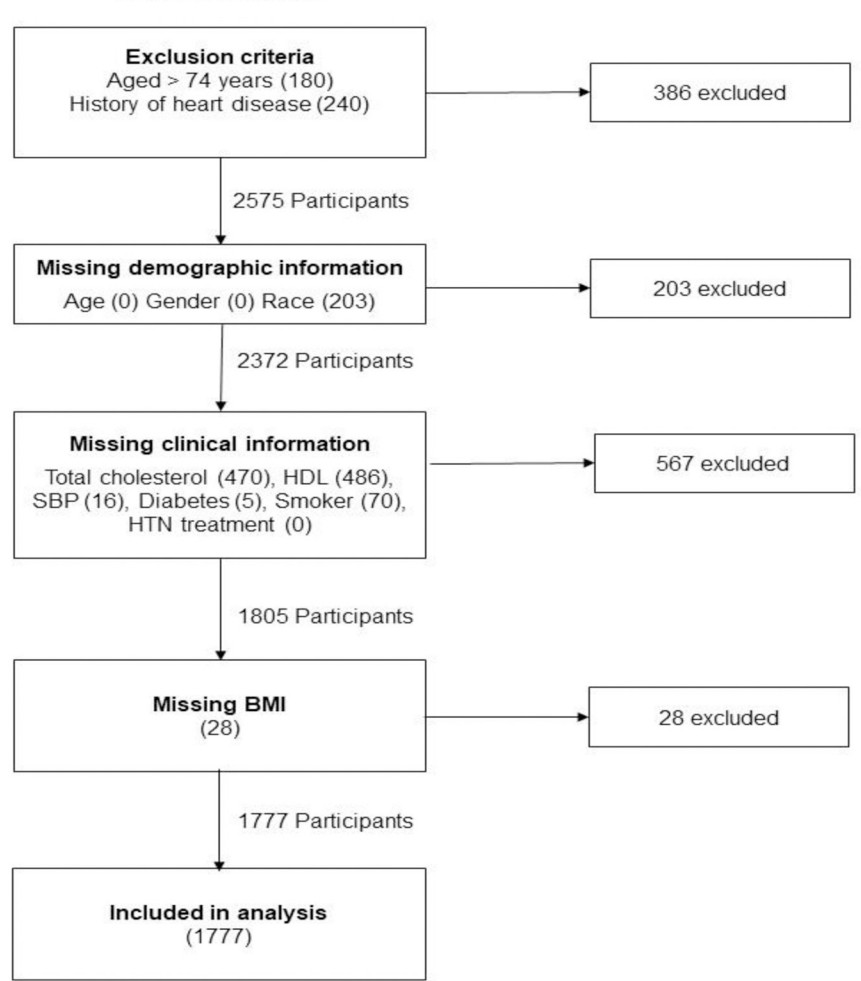

**Fig 1. Flowchart of sample composition.**

## Statistical analyses

We investigated the agreement between the risk scores by cross-tabulating the low/intermediate and high-risk categories based on the different scores. We then calculated kappa statistics for each combination of risk score, taking a value of 0 to <0.01 to indicate no agreement; and 0.01 to 0.20 as none to slight; 0.21 to 0.40 as fair; 0.41 to 0.60 as moderate, 0.61 to 0.80 as substantial; and 0.81 to 1.00 as almost perfect agreement [28].

We explored whether using different risk scores affected the relationship between high CVD risk and the aforementioned SDoH. We used multivariable logistic regression models to compare differences in the association between selected social determinant variables (educational level, occupational grade, household food security, and perceived social status) and high CVD risk according to the five instruments. We examined each SDoH determinant separately in models controlled for age and gender. We did not examine them together in one model as we wanted to identify their direct relationships with the outcome variable without the influence of other predictors, in order to clarify the strength and direction of each SDoH variable's effect.

## Results

Out of 2,961 participants, 386 were excluded due to their age or a previous diagnosis of CVD and a further 798 due to missing data points (Fig 1). The characteristics of the remaining 1,777 participants are shown in Table 2. The mean age of the sample was 55 years. Mean blood pressure was slightly lower in women compared with men (2 mm Hg difference for systolic and 5 for diastolic); however, more women were taking antihypertensive medication (35% vs 25%). Mean BMI, total cholesterol, and HDL cholesterol were all higher in women, whereas the prevalence of smoking and diabetes were higher in men. Around half the sample reported their racial or ethnic background to be Black or Afro-Caribbean. Sensitivity analysis comparing baseline characteristics of included and excluded participants demonstrates slight differences in age and level of education. Mean age (95% CI) of the sample used in this analysis was 55.4 (55.0, 55.8) years, compared with 57.3 (56.9, 57.7) years in the full sample. Our analysis sample had higher educational attainment, with 30.6% (95% CI: 28.4, 32.8) not finishing high school and 22.5 (20.6, 24.5) completing a college degree, compared with 36.1 (95% CI: 34.4, 37.9) and 18.3 (16.9, 19.7), respectively (S1 Table) in the full sample.

Table 3 shows mean 10-year estimated CVD risk scores for each of the five instruments, stratified by gender and age group. For all but one of the sub-groups, Framingham non-lab yielded the largest estimate of 10-year risk across the sample, followed by Framingham lab, ASCVD, WHO lab, and WHO non-lab. For the 40–49 year-old age group, the WHO non-lab estimate was higher than the WHO lab estimate. Mean estimated 10-year risk in the sample was over 2.5 times higher using Framingham non-lab vs WHO non-lab (17.3% vs 6.6%). Across all instruments, estimated 10-year CVD risk was higher in men vs women and increased with age. The sex difference was the greatest using Framingham non-lab (difference of 9.2 percentage points) and the least using the WHO non-lab (1.6 percentage points).

Framingham non-lab categorized the greatest percentage as high risk, followed by Framingham lab, ASCVD, WHO lab, and WHO non-lab. Framingham non-lab was 15 times more

**Table 2.  Baseline CVD risk factors (demographic, physical, and laboratory characteristics) of the ECHORN cohort.**

| Characteristics | Men | Women |
|---|---|---|
| | (N = 635) | (N = 1142) |
| Age, mean (SD) | 55.4 (8.8) | 55.4 (8.9) |
| Blood pressure, mean (SD), mm Hg | | |
| Systolic | 137.0 (18.3) | 135.0 (21.7) |
| Diastolic | 83.6 (10.8) | 78.7 (11.2) |
| Antihypertensive treatment, No. (%) | 162 (25.3) | 398 (34.5) |
| Body mass index, mean (SD) | 27.8 (5.3) | 30.1 (6.4) |
| Total cholesterol, mean (SD), mg/dL[mmol/L] | 187.7 (37.2) | 196.6 (40.0) |
| HDL cholesterol, mean (SD), mg/dL[mmol/L] | 48.0 (13.7) | 55.4 (14.5) |
| Current smoking, No. (%) | 78 (12.2) | 75 (6.5) |
| Diabetes mellitus, No. (%) | 145 (22.7) | 232 (20.1) |
| Race or ethnicity, No. (%) | | |
| White | 48 (7.5) | 73 (6.3) |
| Black/Afro-Caribbean | 323 (50.5) | 585 (50.7) |
| East Indian | 49 (7.7) | 85 (7.4) |
| Hispanic/Latino | 41 (6.4) | 99 (8.6) |
| Puerto Rican/Boricua | 52 (8.1) | 93 (8.1) |
| Mixed | 123 (19.2) | 211 (18.3) |
| Other | 4 (0.6) | 7 (0.6) |

**Table 3. Baseline 10-year CVD risk in the ECHORN cohort according to five different risk estimator tools.**

| | Framingham Simplified | | Framingham General | | ASCVD | | WHO general | | WHO simplified | |
|---|---|---|---|---|---|---|---|---|---|---|
| | Mean | 95% CI | Mean | 95% CI | Mean | 95% CI | Mean | 95% CI | Mean | 95% CI |
| Overall | 17.5 | (16.9,18.1) | 14.0 | (13.5,14.6) | 9.4 | (9,9.9) | 7.1 | (6.9,7.4) | 6.7 | (6.5,6.9) |
| Men | 23.8 | (22.6,24.9) | 19.3 | (18.2,20.4) | 12.3 | (11.5,13.1) | 8.1 | (7.7,8.6) | 7.8 | (7.4,8.1) |
| Women | 14.4 | (13.7,15) | 11.2 | (10.6,11.7) | 7.9 | (7.3,8.4) | 6.5 | (6.2,6.8) | 6.1 | (5.9,6.3) |
| Age (40–49) | 7.3 | (6.9,7.7) | 5.9 | (5.5,6.3) | 2.9 | (2.6,3.3) | 2.8 | (2.6,2.9) | 2.9 | (2.7,3) |
| Age (50–59) | 15.1 | (14.4,15.8) | 12.4 | (11.7,13.2) | 7.4 | (6.8,8) | 5.8 | (5.5,6) | 5.4 | (5.2,5.5) |
| Age (60–69) | 26.8 | (25.5,28) | 21.0 | (19.8,22.2) | 15.1 | (14.2,16) | 10.8 | (10.4,11.2) | 10.0 | (9.7,10.3) |
| Age (70+) | 36.0 | (33.2,38.7) | 27.2 | (24.5,29.9) | 25.7 | (23.4,28) | 17.1 | (16.2,18) | 15.9 | (15.3,16.5) |

Note: CVD risk estimator (variables included in the estimation)

• Framingham non-lab (sex, age, smoking, diabetes, systolic BP, BP treatment, BMI)

• Framingham lab (sex, age, smoking, diabetes, systolic BP, BP treatment, total cholesterol, HDL cholesterol)

• ASCVD (sex, age, race, smoking, diabetes, systolic BP, BP treatment, total cholesterol, HDL cholesterol)

• WHO lab (sex, age, smoking, diabetes, systolic BP, Total cholesterol)

• WHO non-lab (sex, age, smoking, systolic BP, BMI)

likely to classify participants as high risk compared with WHO non-lab (31% vs 2%). The tools produced more similar estimates for the intermediate categories: the highest estimate (Framingham non-lab: 30%) was nine percentage points higher than the lowest (WHO non-lab: 21%) (Fig 2 and S2 Table).

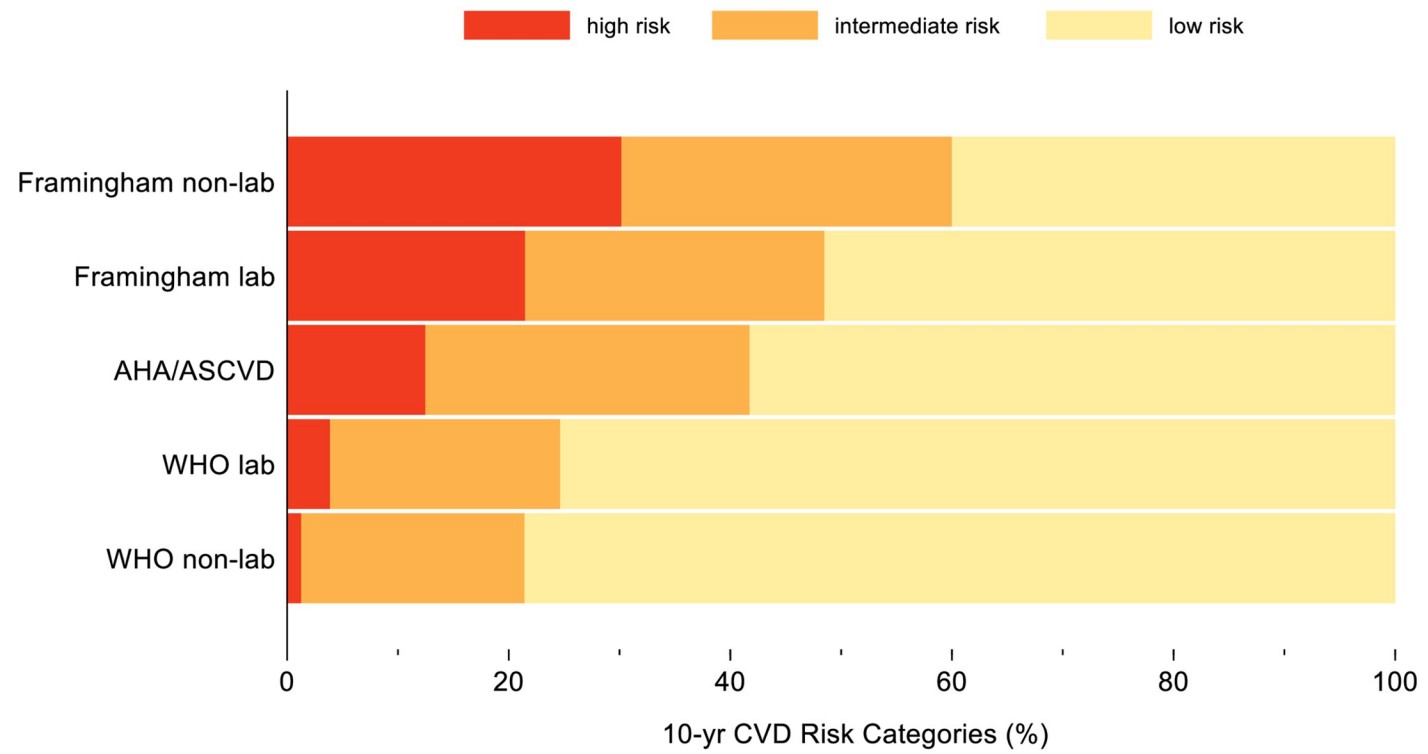

**Fig 2. Bar chart showing 10-year CVD risk categorization in the ECHORN cohort according to five different risk estimator tools.**

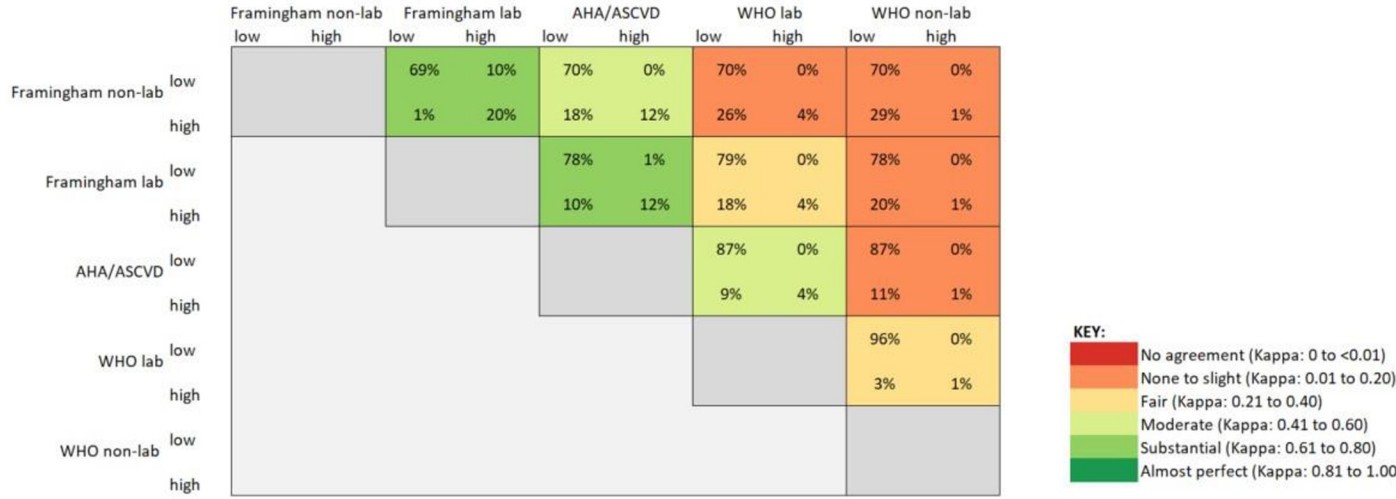

**Fig 3. Agreement between high-risk categories of five cardiovascular risk estimator tools in the ECHORN cohort.**

In terms of agreement between high-risk categories for each of the five instruments, Framingham lab had substantial agreement with Framingham non-lab and ASCVD, with Kappa statistics (95% CI) of 0.72 (0.68, 0.75) and 0.63 (0.58, 0.67), respectively. WHO non-lab showed no agreement or only slight agreement with all other algorithms, except for the WHO lab with which it showed fair agreement (Kappa: 0.34; 95%CI: 0.22, 0.45). All other algorithm combinations showed fair to moderate agreement (Fig 3 and S3 Table).

Table 4 shows the odds of having high estimated 10-year CVD risk according to selected sociodemographic variables, while controlling for age and sex, using each of the five instruments. Using Framingham non-lab and Framingham lab, living in a household with moderate/severe food insecurity was associated with increased odds of high estimated CVD risk (OR = 1.65 and 1.89, respectively), compared with those who live in a food-secure household. Less than high school education was associated with higher estimated CVD risk compared with college graduates (OR range 0.18–0.51) using all instruments except WHO non-lab. Non-professional occupation was associated with higher estimated CVD risk (OR range 1.45–2.09) compared with professional occupation using all instruments except Framingham lab and WHO non-lab. Using WHO non-lab, the odds of high estimated CVD risk did not vary between categories for any of the sociodemographic variables examined.

## Discussion

In this study, we sought to determine the range of estimated 10-year CVD risk across five widely used risk estimate instruments in an ethnically diverse, multi-island Caribbean cohort with a high prevalence of CVD risk factors. Additionally, we sought to determine associations with selected SDoH that have strong evidence-based associations with CVD risk. We found that 10-year CVD risk was nearly three times higher for Framingham non-lab than WHO non-lab. Framingham non-lab categorized the greatest percentage of participants as high risk compared to all the other tools. It categorized 15 times more individuals as high risk than WHO non-lab. The WHO tools had the lowest level of agreement with the other tools we studied while ASCVD had the highest. We also found that moderate food insecurity, lower educational attainment, and non-professional occupation were associated with CVD risk score.

**Table 4. Odds of high estimated 10-year cardiovascular disease risk using 5 different risk estimator tools, by selected social determinants of health.**

| | Prevalence | | Framingham non-lab | | Framingham lab | | ASCVD | | WHO lab | | WHO non-lab | |
|---|---|---|---|---|---|---|---|---|---|---|---|---|
| | N | (%) | OR | 95% CI | OR | 95% CI | OR | 95% CI | OR | 95% CI | OR | 95% CI |
| **Household food security** | | | | | | | | | | | | |
| Food secure | 1,351 | 76 | Reference | | Reference | | Reference | | Reference | | Reference | |
| Mild food insecurity | 277 | 16 | 1.40 | (0.94, 2.00) | 1.29 | (0.95, 2.03) | 1.02 | (0.62, 1.73) | 0.94 | (0.38, 2.33) | 0.78 | (0.22, 2.78) |
| Moderate/severe food insecurity | 149 | 8 | 1.65 | **(1.01, 2.71)** | 1.89 | **(1.12, 3.18)** | 1.15 | (0.53, 2.36) | 0.94 | (0.21, 4.18) | 1.03 | (0.13, 8.39) |
| **Perceived social status** | | | | | | | | | | | | |
| Low social status (tertile 1: score 1–4) | 749 | 44 | Reference | | Reference | | Reference | | Reference | | Reference | |
| Moderate social status (tertile 2: score 5) | 531 | 31 | 0.84 | (0.62, 1.06) | 0.82 | (0.56, 1.06) | 0.94 | (0.64, 1.43) | 1.11 | (0.58, 2.14) | 1.04 | (0.33, 3.25) |
| High social status (tertile 3: 6–10) | 439 | 26 | 1.00 | (0.73, 1.28) | 0.82 | (0.55, 1.05) | 0.93 | (0.60, 1.38) | 0.89 | (0.46, 1.56) | 0.63 | (0.17, 1.21) |
| **Education** | | | | | | | | | | | | |
| Less than high school | 531 | 31 | Reference | | Reference | | Reference | | Reference | | Reference | |
| High school graduate | 402 | 23 | 0.64 | **(0.46, 0.84)** | 0.74 | (0.49, 1.03) | 0.73 | (0.45, 1.14) | 0.55 | (0.24, 1.01) | 0.20 | (0.02, 1.70) |
| Associate degree/ Some college | 413 | 24 | 0.90 | (0.58, 1.06) | 1.13 | (0.78, 1.55) | 0.74 | (0.50, 1.22) | 0.86 | (0.56, 1.76) | 1.55 | (0.50, 4.81) |
| College degree | 391 | 23 | 0.51 | **(0.33, 0.65)** | 0.50 | **(0.34, 0.74)** | 0.48 | **(0.29, 0.78)** | 0.18 | **(0.06, 0.54)** | 0.38 | (0.10, 1.95) |
| **Occupational group** | | | | | | | | | | | | |
| Professional | 430 | 34 | Reference | | Reference | | Reference | | Reference | | Reference | |
| Semi-professional | 621 | 49 | 1.16 | (0.79, 1.44) | 1.34 | (0.97, 1.97) | 1.47 | (0.95, 2.34) | 1.12 | (0.52, 2.40) | 0.28 | (0.05, 1.59) |
| Non-professional | 223 | 18 | 1.45 | **(1.13, 2.38)** | 1.37 | (0.91, 2.24) | 1.79 | **(1.01, 3.17)** | 2.09 | **(1.20, 5.04)** | 3.53 | (0.88, 14.19) |

Ours is among the first published studies to use the 2019 WHO CVD risk estimate instruments [29–32]. It is the first to compare the WHO instrument performance against other instruments and the first to use the tool in a Caribbean population. The 2019 WHO lab and non-lab instruments used one Caribbean cohort, from Puerto Rico, for model derivation. There were no Caribbean studies used for external validation, which included the Asia Pacific Cohort Studies Collaboration (APCSC), the New Zealand primary care-based PREDICT cardiovascular disease cohort (PREDICT-CVD), the Chinese Multi-Provincial Cohort Study, the Health Checks Ubon Ratchathani Study in Thailand, the Tehran Lipids and Glucose Study, and UK Biobank. Following external validation, models were recalibrated using risk factor prevalence data from cross-sectional national surveys (WHO STEPwise approach to non-communicable disease risk factor surveillance) and this included multiple Caribbean nations- Barbados, Dominica, Grenada, St Lucia, and Trinidad and Tobago. The fact that few cohorts were used for model derivation reflects the paucity of Caribbean-specific cohort data, a gap which ECS helps to fill [33, 34].

Our results showed that performance of the Framingham and WHO tools were the most divergent, in that Framingham non-lab predicted a much larger percentage of high-risk participants compared to the WHO non-lab tool. There was strong agreement between WHO non-lab and WHO lab, which was consistent with previous studies that compared WHO risk calculators [29]. It is important to note that the Framingham risk tool has been shown to overestimate CVD risk in different populations, including Chinese and Korean populations [35–38]. This suggests the potential need to recalibrate Framingham for Caribbean-specific contexts. While few studies have been conducted in Caribbean populations, there is some evidence that suggests Framingham may underestimate CVD risk in Caribbean populations [10]. It is important to note the implications of including or excluding specific variables from these instruments, in regard to regional differences in CVD risk factors. For example, the rate of diabetes is high in the Caribbean [2], diabetes is an independent risk factor for CVD, and WHO

non-lab excludes diabetes as a variable. We plan to use longitudinal follow-up data from ECS, once available, to compare these estimations with actual CVD events from the cohort. Such external validation of these existing CVD risk estimate instruments would be an important contribution to the growing evidence base on contextualized approaches to population-level CVD risk estimation [39]. One analytic approach to addressing low concordance between instruments would be combination modeling, such as COVID-19 ensemble modeling. Evidence suggests that such models can provide "reliable and comparatively accurate forecast[s] that exceed the performance of. . .the models that contribute to it" [40]. Discordance between risk instruments has real implications at the individual and population levels, especially in resource constrained settings. Individuals identified as having high CVD risk will likely be prescribed more medicines and have closer recommended follow up, for example. On the other hand, those who experience a cardiovascular event will also require increased resources. At the individual level, this translates into higher out of pocket expenditures, lost time at work, and greater travel, while population-level implications include increased medicine demand and greater primary care or specialist resources.

In addition to the aforementioned discussion of inclusion or exclusion of specific variables, the role of race in these instruments is critical to address herein. Of our selected instruments, only ASCVD includes a race variable and this is either "African American" or "other". This dichotomous variable is surely an oversimplifaction of a complex and controversial social construct. In a recent systematic review of 363 CVD risk estimate instruments, most of which were developed in North American and European contexts, approximately 10% included a race variable [39]. The inclusion of race in medical risk estimate tools has been seriously questioned, given the threat of worsening already entrenched health disparities. One example is in regard to estimation of kidney function and its impact on kidney transplantation [41, 42]. This same debate has taken hold in the realm of CVD risk estimate instruments. The goal is the development of race- and ethnicity-free CVD risk estimate instruments that "obviate race-associated risk misestimation and racializing treatment practices, and instead incorporate measures of SDoH that mediate race associated risk differences" [43]. Two recent prominent publications deserve special attention within this context. In one, using contemporary data from the US-based biracial REGARDS cohort study, investigators found that the removal of race and the addition of SDoH neither improved nor worsened performance of the ASCVD pooled cohort equation. The SDoH incorporated in this model included: SES, urban versus rural residence, measures of social support and racial segregation, health insurance coverage, and area-level variables incuding residing in an area of health professional shortage and a measure of poor public health infrastructure [17]. In the other, the American Heart Association released an updated version of the ASCVD instrument. This new instrument, Predicting Risk of CVD EVENTs (PREVENT) builds upon ASCVD in multiple important ways: it uses contemporary data from over 6 million racially diverse US adults, lowers the starting eligible age to 30 years, provides both 10- and 30-year estimated risk for heart failure, stroke/myocardial infarction (ASCVD), and composite CVD, removes the race variable, and acknowledges SDoH by incorporating a social deprivation index (a measure of social disadvantage based on ZIP code [18]. Given these important changes to ASCVD, future comparative analyses similar to ours certainly will need to include PREVENT. That said, given PREVENT's use of ZIP code, it is not useable in non-US populations, in its current form. Finally, it is important to note the potential for racial disparities in the real-world application of clinical tools, when biological differences between groups is ignored. One such example is the large difference in occult hypoxemia, undetected by pulse oximeters, among Black compared to White patients and the implications of this in the COVID-19 pandemic.

Our examination of SDoH and CVD risk highlighted food insecurity, those with the lowest level compared to the highest level of education, and non-professional occupation. Our results showed positive associations between these SDoH and CVD risk, though the associations were inconsisent across risk instruments (e.g. food insecurity was associated with risk only using Framingham lab and non-lab) and the risk score hierarchy is less strong once SDoH were introduced. These findings add support to prior calls for the consideration of SDoH in CVD risk calculators [12, 44]. Our findings are similar to other studies that also demonstrated that food insecurity was associated with increased CVD risk and CVD mortality [45, 46]. Studies have also suggested the consideration of educational attainment in CVD risk calculators. Evidence has long shown an inverse relationship between educational attainment and CVD incidence and mortality. However, like food insecurity, educational attainment is not considered in CVD risk scores. A 2009 study examined the potential impact of incorporating educational attainment in Framingham found that its inclusion improved risk prediction [47]. Our results also indicated that non-professional occupations were associated with higher estimated CVD risk. In a study of Japanese adults, professional occupations typically yielded fewer CVD events than non-professional occupations [48]. An important consideration is also the effective working hours and income associated with non-professional or professional occupations, factors that would adversely or positively impact CVD risk. Our findings add empirically to prior evidence suggesting a potential benefit to acknowledging SDoH in CVD risk prediction instruments, especially for ethnically diverse populations. An important practical problem would be the need for a global movement towards standardization of SDoH variables. Efforts such as the WHO Commission on the SDoH [49] and the Centers for Disease Control and Prevention Healthy People 2020 framework [50] represent such progress.

In the context of our findings, this study has several limitations. First, we only used five CVD risk tools in this study. We selected widely used tools with applicability in lower resource settings and selected only five so to allow for simpler comparisons. Other tools may have performed differently in our analyses. Second, our data did have a fair amount of missingness. However, our sensitivity analyses demonstrated that these missing data did notchange the demographic profile of our sample relative to the full cohort. That said, though the sensitivity analysis suggests that full and partial samples had certain similarities, the missingness limits the strength of our findings. Third, we selected few SDoH variables, those which are most commonly cited in the literature as having CVD risk associations. However, other SDoH could have generated further insights into possible associations with CVD risk.

## Conclusion

Improved CVD risk tools, potentially incorporating key SDOH, stand to benefit both individual-level care and population-level preventive messaging and public health. Hard CVD outcomes data from ECS and other multi-ethnic cohort studies will contribute to strengthening the predictive power of CVD risk tools.

## Supporting information

**S1 Table. Sensitivity analysis showing demographic characteristics of full sample (n = 2575) vs those with complete data (n = 1777).** Values highlighted in bold signify a statistically significant difference at the 5% level between the full sample and study sample. (DOCX)

**S2 Table. Baseline 10-year CVD risk categorization in the ECHORN cohort according to five different risk estimator tools.**
(DOCX)

**S3 Table. Agreement between high-risk categories of five cardiovascular risk estimator tools in the ECHORN cohort.**
(DOCX)

## Acknowledgments

### AI statement

During the preparation of this work, JIS used ChatGPT 3.5 in order to draft the Abstract and Research in Context sections of this manuscript. After using this tool, all authors reviewed and edited the content as needed and take full responsibility for the content of the publication.

## Author Contributions

**Conceptualization:** Jeremy I. Schwartz, Carol Oladele.

**Data curation:** Christina Howitt.

**Formal analysis:** Christina Howitt.

**Funding acquisition:** Oswald P. Adams, Rohan G. Maharaj, Cruz M. Nazario, Maxine Nunez, Marcella Nunez-Smith.

**Investigation:** Sanya Nair.

**Methodology:** Jeremy I. Schwartz, Carol Oladele.

**Writing – original draft:** Jeremy I. Schwartz, Christina Howitt.

**Writing – review & editing:** Jeremy I. Schwartz, Christina Howitt, Sumitha Raman, Sanya Nair, Saria Hassan, Carol Oladele, Ian R. Hambleton, Daniel F. Sarpong, Oswald P. Adams, Rohan G. Maharaj, Cruz M. Nazario, Maxine Nunez, Marcella Nunez-Smith.

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
