## [Decision Letter · Decision Letter 0]

7 Nov 2024

PONE-D-24-23025Assessing cardiovascular disease risk and social determinants of health: a comparative analysis of five risk estimation instruments using data from the Eastern Caribbean Health Outcomes Research NetworkPLOS ONE

Dear Dr. Schwartz,

Thank you for submitting your manuscript to PLOS ONE. After careful consideration, we feel that it has merit but does not fully meet PLOS ONE’s publication criteria as it currently stands. Therefore, we invite you to submit a revised version of the manuscript that addresses the points raised during the review process. Read carefully the comments of reviewers and answer them accordingly.

We look forward to receiving your revised manuscript.

Kind regards,

Paulo Alexandre Azevedo Pereira Santos, PhD

Academic Editor

PLOS ONE

“NIH grants U24MD006938, U54MD010711, K23HL152368, UL1TR000142, and Yale School of Medicine.”

Additional Editor Comments:

Thank you for your submission. Please, read carefully the comments of reviewers and answer them accordingly.

Reviewers' comments:

Reviewer's Responses to Questions

**Comments to the Author**

1. Is the manuscript technically sound, and do the data support the conclusions?

Reviewer #1: Yes

Reviewer #2: Partly

2. Has the statistical analysis been performed appropriately and rigorously? 

Reviewer #1: I Don't Know

Reviewer #2: Yes

3. Have the authors made all data underlying the findings in their manuscript fully available?

Reviewer #1: Yes

Reviewer #2: Yes

4. Is the manuscript presented in an intelligible fashion and written in standard English?

Reviewer #1: Yes

Reviewer #2: Yes

5. Review Comments to the Author

Reviewer #1: This is a well-written article comparing cardiovascular risk assessment tools in the Eastern Caribbean Health Outcomes Research Network Cohort Study. This study further assesses the association of social determinants of health with high cardiovascular risk. Some general questions and thoughts are as immediately below followed by line comments. I wonder if this would be best suited for PLOS ONE or PLOS Global Public Health? The article will need statistical review.

General questions and thoughts:

--Are the "significant discrepancies in CVD risk assessment across tools" similarly discrepant in other cohorts or more/less discrepant in the ECHORN cohort?

--Table 4, Lines 401-402: I admittedly go back and forth with regards to wanting an analysis similar to Table 4 between participants who identified as Black and Other (including White in Other here given lower n numbers).. Or having a model that includes race/ethnicity with the age and sex adjustments if statistically appropriate. It may be somewhat unaligned with the described goal to develop "race- and ethnicity-free CVD risk estimates" but it also seems important to understand and interpret similarities or differences. I defer to the authors' expertise and preferences here.

--To note, BMI also has limitations in assessing cardiovascular risk in diverse populations (https://pubmed.ncbi.nlm.nih.gov/26149446/) and was curious how BMI categories here align with cardiovascular risk, understanding however that the focus here is on SDoH and that this can be a future study. Studies of cardiovascular health (rather than CVD risk) could also be a future study as it seems AHA Life's Simple 7 scores (but perhaps not Life's Essential 8) can be calculated from the ECHORN data..

Line comments:

--Line 77, Line 331, Line 406: what determines "strong association" since food insecurity was associated with higher CVD risk in two out of the 5 risk tools (and high school graduate alone was associated in lower CVD risk for one out of the five tools)?

--Line 78, Lines 405-406: did analyses really show association between lower educational attainment and increased CVD risk? or rather, i believe from Table 4, lower odds of high CVD risk for people with a college degree (compared to people with less than high school education)? i understand that these are similar, but they do not seem quite the same (especially in the relative absence of differences compared to reference group for the high school graduate and associate degree/some college groups).

--Line 123: is this e.g. necessary or appropriately placed?

--Lines 134-135: assuming these are both supposed to be em dashes, they appear to be different..

--Line 147: semi-permanent (no space needed)

--Line 154: audio-assisted (no space needed)

--Line 157: A1c

--Line 159: capitalize to align with other section headings

--Line 161: no need for comma

--Line 174: is there a reason why A1c >=6.5% was not included here (since included in measurements per line 157)?

--Line 195: Is there a reason why Globorisk (or Globorisk-LAC if available, https://pubmed.ncbi.nlm.nih.gov/35711683/) was not included? somewhat answered in Lines 428-430..

--Lines 199, Line 202, Lines 376-378: It includes White, African American, and Other, correct?

--Lines 207-208: why were participants who selected white classified as other and not white?

--Line 212: tools?

--Line 234: de-identified (no space needed)

--Line 261: suggest adding "in the full sample" to be clear

--Table 2: I'm assuming the total cholesterol and HDL were appropriately converted between mg/dL and mmol/L for the respective CVD risk tools?

--Table 3, Table 4, Supplementary Table 3: suggest ASCVD rather than AHA/ASCVD

--Table 4: To my understanding, income relative to population was not measured rather perceived social status or subjective socioeconomic status. Advised changing the references to income here.

--Table 4: suggest associate or associate's

--Line 333: https://pubmed.ncbi.nlm.nih.gov/37648706/ compared the WHO instrument performance against another instrument, correct?

--Lines 345-346: suggest a clearer and simpler word that disclaim and adding a reference

--Line 364: suggest COVID-19

--Line 399: ZIP

--Lines 399-400: including PREVENT (as it currently is) for analyses in non-US settings will not be possible since based on U.S. Postal Service ZIP code (and associated deprivation indices) though, correct? This sentence could be removed if desired or adapted.

--Line 447: I don't see a "Research in Context" section?

Reviewer #2: There are some minor comments:

- Line 227: The authors state that in their sensitivity analysis they compare demographics and outcomes among included and excluded participants. But the Supplementary Table 1 only includes data on some demographics and they do not include any data about outcomes of the study. And in line 432 they state that their sensitivity analysis demonstrate that missing data did not affect the risk score. As this is hard to believe (if they do not have data about these participants, they cannot calculate the risk scores, and unless they provide data on the incidence of Cardiovascular events or on disease prevalence among excluded individuals) they cannot assume that the sesnsitivity analysis discards bias in this study.

Methodology: It would be better if the authors describe more thoroughly the multivariable regression models they use (at least in the supplementary material) and the results of these models.

In the results displayed on table 4, there is a remarkable great range on the CI in the WHO non-lab model. This aspect deserves some explanation in the Discussion, as ther is no such result in the Framingham non-lab model, despite they include roughly the same variables (except Blood-pressure treatment and diabetes).

In line 355 the authors state thate reference 10 suggest that Framngham may underestimate CVD in Caribbean population, when they should speak about CVD risk.

6. PLOS authors have the option to publish the peer review history of their article (what does this mean?). If published, this will include your full peer review and any attached files.

Reviewer #1: No

Reviewer #2: No

---

## [Author Response · Author response to Decision Letter 0]

27 Nov 2024

November 25, 2024

Dear PLOS ONE editorial staff, 

We appreciate the thoughtful review of our manuscript. Herein, please find our rebuttals to each reviewer comment, in italics.

Sincerely,

Jeremy Schwartz

Reviewer #1

1. Are the "significant discrepancies in CVD risk assessment across tools" similarly discrepant in other cohorts or more/less discrepant in the ECHORN cohort?

As we write in lines 105-107, these different risk assessment tools consistently demonstrate discrepancies. In this study, we did not seek to quantify the degree of discrepancy in ECHORN when compared to other cohorts.

2. Table 4, Lines 401-402: I admittedly go back and forth with regards to wanting an analysis similar to Table 4 between participants who identified as Black and Other (including White in Other here given lower n numbers).. Or having a model that includes race/ethnicity with the age and sex adjustments if statistically appropriate. It may be somewhat unaligned with the described goal to develop "race- and ethnicity-free CVD risk estimates" but it also seems important to understand and interpret similarities or differences. I defer to the authors' expertise and preferences here.

We appreciate the tension voiced by the reviewer, who is deferring to our expertise and preferences. First, it is difficult to define/assign race/ethnicity in multi-ethnic populations such as the ECHORN cohort, where mixed ancestry is common. Second, in this study, we chose to focus on SDoH rather than race as these directly influence lifestyle behaviors, stress levels, and access to preventative care, all of which contribute significantly to CVD risk. Race and ethnicity often act as proxies for these underlying inequities but do not capture the specific mechanisms driving risk. 

3. To note, BMI also has limitations in assessing cardiovascular risk in diverse populations (https://pubmed.ncbi.nlm.nih.gov/26149446/) and was curious how BMI categories here align with cardiovascular risk, understanding however that the focus here is on SDoH and that this can be a future study. Studies of cardiovascular health (rather than CVD risk) could also be a future study as it seems AHA Life's Simple 7 scores (but perhaps not Life's Essential 8) can be calculated from the ECHORN data.

Our colleagues have previously published an analysis of the relationship between BMI and CVD risk in the ECHORN cohort: https://pubmed.ncbi.nlm.nih.gov/33632164/.

4. Line 77, Line 331, Line 406: what determines "strong association" since food insecurity was associated with higher CVD risk in two out of the 5 risk tools (and high school graduate alone was associated in lower CVD risk for one out of the five tools)?

We agree that this is better described as an association, rather than “strong”. We have removed the word “strong” from the text

5. Line 78, Lines 405-406: did analyses really show association between lower educational attainment and increased CVD risk? or rather, i believe from Table 4, lower odds of high CVD risk for people with a college degree (compared to people with less than high school education)? i understand that these are similar, but they do not seem quite the same (especially in the relative absence of differences compared to reference group for the high school graduate and associate degree/some college groups).

We agree and have amended the text so that it is clear that we are referring to the highest compared to the lowest level of education, rather than a gradient across all levels. 

6. Line 123: is this e.g. necessary or appropriately placed?

Removed

7. Lines 134-135: assuming these are both supposed to be em dashes, they appear to be different.

Changed

8. Line 147: semi-permanent (no space needed)

There is not a space- unclear what the reviewer is referring to

9. Line 154: audio-assisted (no space needed)

There is not a space- unclear what the reviewer is referring to

10. Line 157: A1c

Removed altogether, as hemoglobin A1c was not part of the current analysis.

11. Line 159: capitalize to align with other section headings

Already is capitalized.

12. Line 161: no need for comma

Removed comma

13. Line 174: is there a reason why A1c >=6.5% was not included here (since included in measurements per line 157)?

We excluded A1c to bring our definition of diabetes in line with those used in population-based surveys in the US (NHANES) and globally (WHO STEPS). Agreed that it is confusing to mention A2c when we did not use it, so we have removed reference to it as per above. 

14. Line 195: Is there a reason why Globorisk (or Globorisk-LAC if available, https://pubmed.ncbi.nlm.nih.gov/35711683/) was not included? somewhat answered in Lines 428-430..

As noted by the reviewer, we do refer to Globorisk. However, we could not be exhaustive in our use of CVD risk calculators for practical reasons. We selected the included calculators based on informal conversations with local clinicians, who identified Framingham and ASCVD as the most commonly used calculators in clinical practice. We included the WHO tool due to its calibration using risk factor data from Caribbean populations, and the interest generated in it by local clinicians due to promotion of the tool in the region by WHO. The Globorisk-LAC is less well-known, and although it was derived using one Caribbean cohort, the overwhelming majority of the sample was from Latin America (only 3% of the sample is Caribbean).

15. Lines 199, Line 202, Lines 376-378: It includes White, African American, and Other, correct?

Please see response to next comment.

16. Lines 207-208: why were participants who selected white classified as other and not white?

The ASCVD tools is the only one from those selected for this study that requires race to be entered. It either designates participants as “African American” or “Other”. Note that the online tool allows one to select “White” also, but the underlying algorithm groups “White” and “Other” together. We have explained this in lines 205-206 (new line numbering after editing).

17. Line 212: tools?

Edited accordingly.

18. Line 234: de-identified (no space needed)

Edited accordingly.

19. Line 261: suggest adding "in the full sample" to be clear

Edited accordingly.

20. Table 2: I'm assuming the total cholesterol and HDL were appropriately converted between mg/dL and mmol/L for the respective CVD risk tools?

We confirm that we entered the measurements into the tools in the appropriate units.

21. Table 3, Table 4, Supplementary Table 3: suggest ASCVD rather than AHA/ASCVD

We have changed the text to ASCVD.

21. Table 4: To my understanding, income relative to population was not measured rather perceived social status or subjective socioeconomic status. Advised changing the references to income here.

This is our error and we have edited Table 4 accordingly. 

22. Table 4: suggest associate or associate's

Edited accordingly.

23. Line 333: https://pubmed.ncbi.nlm.nih.gov/37648706/ compared the WHO instrument performance against another instrument, correct?

Correct. This is one reference we have not included. We are not claiming that our paper is the first to use the new WHO instrument- but among the first.

24. Lines 345-346: suggest a clearer and simpler word that disclaim and adding a reference

On reconsideration, given conflicting data literature, we decided to remove this sentence altogether.

25. Line 364: suggest COVID-19

Edited accordingly.

26. Line 399: ZIP

Edited accordingly.

27. Lines 399-400: including PREVENT (as it currently is) for analyses in non-US settings will not be possible since based on U.S. Postal Service ZIP code (and associated deprivation indices) though, correct? This sentence could be removed if desired or adapted.

We edited to acknowledge this.

28. Line 447: I don't see a "Research in Context" section?

To our understanding, PLOS ONE does not require a Research in Context section. Editors can advise if otherwise.

Reviewer #2

1. Line 227: The authors state that in their sensitivity analysis they compare demographics and outcomes among included and excluded participants. But the Supplementary Table 1 only includes data on some demographics and they do not include any data about outcomes of the study. And in line 432 they state that their sensitivity analysis demonstrate that missing data did not affect the risk score. As this is hard to believe (if they do not have data about these participants, they cannot calculate the risk scores, and unless they provide data on the incidence of Cardiovascular events or on disease prevalence among excluded individuals) they cannot assume that the sensitivity analysis discards bias in this study.

We agree. The text has been changed to state that the sensitivity analyses showed that the missing data did not change the demographic characteristics pf the sample relative to the full cohort. We have removed the reference to comparison of outcomes. We acknowledge in the limitations that missing data may limit the strength of our findings. 

2. Methodology: It would be better if the authors describe more thoroughly the multivariable regression models they use (at least in the supplementary material) and the results of these models.

We have added more information on the modeling strategy to the “Statistical Analyses” section. 

3. In the results displayed on table 4, there is a remarkable great range on the CI in the WHO non-lab model. This aspect deserves some explanation in the Discussion, as there is no such result in the Framingham non-lab model, despite they include roughly the same variables (except Blood-pressure treatment and diabetes).

When the relationship between the independent and dependent variables is weak, the estimate's precision is lower, which can result in wide CIs. Unlike the Framingham non-lab, the WHO non-lab was not associated with any SDoH variable. This likely caused the wide CIs in table 4.

4. In line 355 the authors state that reference 10 suggest that Framingham may underestimate CVD in Caribbean population, when they should speak about CVD risk.

Noted and changed.

---

## [Editor Report · Decision Letter 1]

13 Dec 2024

Assessing cardiovascular disease risk and social determinants of health: a comparative analysis of five risk estimation instruments using data from the Eastern Caribbean Health Outcomes Research Network

PONE-D-24-23025R1

Dear Dr. Schwartz,

We’re pleased to inform you that your manuscript has been judged scientifically suitable for publication and will be formally accepted for publication once it meets all outstanding technical requirements.

Kind regards,

Paulo Santos, PhD

Academic Editor

PLOS ONE

---

## [Editor Report · Acceptance letter]

10 Jan 2025

PONE-D-24-23025R1 

PLOS ONE

Dear Dr. Schwartz, 

I'm pleased to inform you that your manuscript has been deemed suitable for publication in PLOS ONE. Congratulations! Your manuscript is now being handed over to our production team.

Kind regards, 

on behalf of

Professor Paulo Alexandre Azevedo Pereira Santos 

Academic Editor

PLOS ONE